# NONLINEAR MULTIREGION NEURAL DYNAMICS WITH PARAMETRIC IMPULSE RESPONSE COMMUNICATION CHANNELS

**Matthew Dowling & Cristina Savin** [*]
Center for Neural Science
New York University
{md6276,csavin}@nyu.edu

## ABSTRACT

Cognition arises from the coordinated interaction of brain regions with distinct computational roles. Despite improvements in our ability to extract the dynamics underlying circuit computation from population activity recorded in individual areas, understanding how multiple areas jointly support distributed computation remains a challenge. As part of this effort, we propose a multi-region neural dynamics model composed of two building blocks: *i)* within-region (potentially driven) nonlinear dynamics and *ii)* communication channels between regions, parameterized through their impulse response. Together, these choices make it possible to learn nonlinear neural population dynamics and understand the flow of information between regions by drawing from the rich literature of linear systems theory. We develop a state noise inversion free variational filtering and learning algorithm for our model and show, through neuroscientifically inspired numerical experiments, how the proposed model can reveal interpretable characterizations of the local computations within and the flow of information between neural populations. We further validate the efficacy of our approach using simultaneous population recordings from areas V1 and V2.

## 1 INTRODUCTION

Perception, choice and action engage neural circuits distributed across the brain (Chen et al., 2024; Khilkevich et al., 2024; Noel et al., 2024; Pinto et al., 2022; Machado et al., 2022; Ebrahimi et al., 2022). Despite technological advances that facilitate recording from multiple, anatomically distinct, populations of neurons (Steinmetz et al., 2021), understanding neural computation at the level of multiple interacting populations remains a statistical and theoretical challenge. Making progress requires new theoretical frameworks describing how global computations arise from multiple interacting circuits (Perich & Rajan, 2020), each with potentially complex local nonlinear dynamics, and new statistical tools that extract such structure directly from recorded neural activity during behavior.

One prominent set of approaches for measuring interarea interactions based on neural data focus on *communication subspaces* (Semedo et al., 2019). Rather than modeling local circuit dynamics explicitly, these approaches aim to partition population response variability into 'private' dimensions, that are local to an area, and 'shared' dimensions reflecting the flow of information across areas. In its simplest form, this partitioning is formalized as low-rank regression or canonical correlation analysis, for directional or undirectional communication, respectively (Semedo et al., 2020). Additionally, Gaussian Process (GP) priors for the latents enforce temporal regularities, and explicitly model features like communication delays (Gokcen et al., 2022; 2024), frequency and phase delays (Li et al., 2024) or additional task-relevant covariates (Balzani et al., 2023).

Building upon a decade of progress in latent state estimation from neural population activity (Paninski et al., 2010; Cunningham & Yu, 2014; Duncker & Sahani, 2021), other approaches directly model the dynamics within each areas and the interactions between them. The simplest such models use linear dynamical systems (LDS) for capturing within area dynamics. For instance, gLARA (group

---

[*]Center for Data Science, NYU

latent autoregressive analysis) (Semedo et al., 2014) assumes that within- and between- population dynamics are both governed by LDSs. More recently, the state-space representation of finitely differentiable GPs (Li et al., 2024) blurs the distinction between GP-prior based communication subspaces and LDS methods, although these are mainly leveraged for efficient inference. Oftentimes, multi-area approaches can be seen as special cases of single area models, with additional parameter constraints. For instance, Glaser et al. (2020) adapts recurrent switching linear dynamical systems (rSLDS) to construct a multi-population sticky rSLDS (mp-srSLDS) of neural dynamics. This allows for nonlinear within area dynamics and instantaneous linear information flow between them. The most complex multi-area model is MR-SDS (multi-region switching dynamical systems), which uses neural networks to parametrize arbitrary nonlinearities for within area dynamics and across areas communication and uses switching to capture global transitions between such nonlinear systems to model behavioral state (Karniol-Tambour et al., 2022). Closest to the circuit level, multi-region recurrent neural networks (RNNs) can be fit directly to single neuron responses Perich et al. (2020), which provides direct current estimates but leaves understanding the low dimensional dynamical systems structure of the solution to post-hoc investigation. Overall, different approaches provide different trade-offs between flexibility and interpretability (see Appendix Table S1). None of the existing methods fully reflect the nature of distributed computation as formalized in current circuit level theories (Bredenberg et al., 2024; Langdon et al., 2023; Mišić & Sporns, 2016).

Here we develop a probabilistic generative model that accounts for the nonlinear nature of neural dynamics and characterizes communication between regions using channels that are parameterized by their impulse response – blending expressive nonlinear region specific dynamics with interpretable characterizations of the flow of information between regions. Our major methodological contributions include i) the generative model of latent neural dynamics that combines node-specific nonlinear dynamical systems parameterized by deep neural networks with linear communication channels between regions, which we term MRDS-IR (for MultiRegion Dynamical Systems with Impulse Response communication channels) ii) an end-to-end variational methodology, using a state-noise inversion free filtering algorithm, streamlining the treatment of approximate inference in state-space graphical models with hybrid stochastic/deterministic transitions. Through several neuroscientifically inspired numerical experiments including integration, gating of information flow and rhythmic timing, we demonstrate the use of our approach to make sense of the underlying computation behind observed multi-population neural responses. We also show that our approach reveals meaningful features of neural activity in joint population recordings from visual areas V1 and V2.

## 2 MODELING MULTI-AREA NEURAL DYNAMICS DURING BEHAVIOR

### 2.1 BACKGROUND

**State-space models.** State-space graphical models provide a principled framework for data driven learning of neural population dynamics (Paninski et al., 2010). For a single neural population, recorded neural activity, $\mathbf{y}_t \in \mathbb{R}^N$, at time, $t$, is modeled as reflecting a lower-dimensional population latent state, $\mathbf{z}_t \in \mathbb{R}^L$, which evolves as a dynamical system parameterized by $\boldsymbol{\theta}$,

$$\mathbf{z}_t = \mathbf{f}_{\boldsymbol{\theta}}(\mathbf{z}_{t-1}, \mathbf{c}_t) + \mathbf{w}_t \quad \text{(latent process)} \qquad \mathbf{y}_t \mid \mathbf{z}_t \sim p(\mathbf{y}_t \mid \mathbf{z}_t) \quad \text{(observation model)} \qquad (1)$$

where $\mathbf{w}_t \sim \mathcal{N}(\mathbf{0}, \mathbf{Q})$, and $\mathbf{c}_t$ denotes (optional) inputs/stimuli.

Generalizing this formalism to simultaneous recordings from $K$ regions, a natural choice is to partition the latent space into $K$ groups of latent variables, with population responses in any given depending only on the latents of that region (Gokcen et al., 2022; Li et al., 2024; Karniol-Tambour et al., 2022; Semedo et al., 2014), $p(\mathbf{y}_t^{(k)} \mid \mathbf{z}_t^{(1)}, \mathbf{z}_t^{(2)}, \dots, \mathbf{z}_t^{(K)}) = p(\mathbf{y}_t^{(k)} \mid \mathbf{z}_t^{(k)})$, where $\mathbf{y}_t^{(k)}$ and $\mathbf{z}_t^{(k)}$ are the population activity and latent state associated with region $k$, respectively. Within this common structure, different multi-region models make different choices for the functional form of the latent space and the dependencies linking latents across regions. This structure determines not only the model's expressiveness but also its ability to capture crucial aspects of neural population dynamics. One important such feature is the latency in communication between regions, reflecting the time delays inherent in signal propagation, which is absent in most process models (Glaser et al., 2020; Karniol-Tambour et al., 2022) or realized by introducing dependence on a finite state history (Semedo et al., 2014). To provide a more flexible framework for modeling signal propagation between regions, we consider principles from linear system theory.

**Characterizing communication channels via their impulse response.** The general premise of our approach for modeling communication channels is that signal propagation between regions can be well approximated by sufficiently expressive linear filters, allowing for propagation delays and temporal filtering, e.g. preferential transfer of information in a specific frequency band (Bastos et al., 2015), while keeping the model tractable. Two fundamental concepts for understanding a linear system are i) its impulse response and ii) its transfer function; they offer complementary perspectives on the system's input to output map, characterizing information flow in both time and frequency (Kailath, 1980; Chen, 1984; Brockett, 2015). Consider an $N_{\text{in}}$ dimensional input signal, $\mathbf{u}_t$, driving a linear system to produce an $N_{\text{out}}$ dimensional output signal, $\mathbf{x}_t$. The impulse response of the system, $\mathbf{h}_t$, is a $N_{\text{out}} \times N_{\text{in}}$ dimensional matrix, with entry $(i,j)$ given by the $[\mathbf{x}_t]_i$ output when $[\mathbf{u}_t]_j$ is the unit impulse. By superposition, $\mathbf{x}_t$ and $\mathbf{u}_t$ are related by convolution so that,

$$\mathbf{x}_t = \sum_{\tau=-\infty}^{t} \mathbf{h}_{t-\tau} \mathbf{u}_\tau. \tag{2}$$

An alternative characterization, more suited to understanding frequency-domain properties, is the transfer function, which for discrete-time systems is the $\mathcal{Z}$-transform of the impulse response,

$$\mathsf{H}(z) = \sum_{t=-\infty}^{\infty} z^{-t} \mathbf{h}_t \tag{3}$$

where the transfer function, $\mathsf{H}(z)$, is also an $N_{\text{out}} \times N_{\text{in}}$ dimensional matrix whose $(i,j)$ entry characterizes how frequency content changes from input dimension $j$ to output dimension $i$. If $\mathbf{x}(z)$ and $\mathbf{u}(z)$ are the $\mathcal{Z}$-transform of $\mathbf{x}_t$ and $\mathbf{u}_t$ respectively, then in the $\mathcal{Z}$-domain they can be related by $\mathbf{x}(z) = \mathsf{H}(z)\mathbf{u}(z)$. Importantly, if the entries of $\mathsf{H}(z)$ are all rational in $z$ and the degree of the denominator exceeds that of the numerator, then a finite-dimensional *realization* of that system can be implemented by an LDS.[1] This means that for any strictly proper[2] transfer function satisfying those properties, there exists a tuple $(\mathbf{A}, \mathbf{B}, \mathbf{C})$ that parameterize an LDS,

$$\mathbf{x}_t = \mathbf{C}\gamma_t \qquad\qquad \gamma_t = \mathbf{A}\gamma_{t-1} + \mathbf{B}\mathbf{u}_t \tag{4}$$

whose impulse response and transfer function match $\mathbf{h}_t$ and $\mathsf{H}(z)$ respectively, and can be written in terms of the LDS parameters as,

$$\mathbf{h}_t = \mathbf{C}\mathbf{A}^{t-1}\mathbf{B} \qquad\qquad \mathsf{H}(z) = \mathbf{C}(z\mathbf{I} - \mathbf{A})^{-1}\mathbf{B} \tag{5}$$

Consequently, impulse response descriptions of communication channels can be directly incorporated into state-space model descriptions of multi-region neural dynamics. For understanding temporal characteristics such as delays in communication channels, the impulse response can provide an informative description; how information may be attenuated or amplified at different frequencies is better understood through the transfer function.

## 2.2 THE MRDS-IR GENERATIVE MODEL

We consider region specific latent states driven by their own recurrent dynamics subject to filtered content of other region's latent state history, with dynamics of the form,

$$\mathbf{z}_t^{(k)} = \mathbf{f}_k(\mathbf{z}_{t-1}^{(k)}) + \sum_{\ell \neq k} \mathcal{H}_{k,\ell}(\mathbf{z}_{1:t-1}^{(\ell)}) + \mathcal{G}_k(\mathbf{c}_t^{(k)}) + \mathbf{w}_t^{(k)} \tag{6}$$

where $\mathcal{G}_k$ maps region specific stimuli/inputs to the latent space, and $\mathcal{H}_{k,\ell}$ transforms the latent state history of region $\ell$ into an input to region $k$ – acting as a directed and causal[3] *channel* that controls the transmission of information between regions. This state-space structure is mathematically general, with many existing multi-region neural dynamics models in the literature as special cases.

We model channels between regions, $\mathcal{H}_{k,\ell}$, as linear filters parameterized by their impulse response, as explained above, which allows us to build a fully Markovian representation in a higher dimensional state-space (Åström & Wittenmark, 2013). We structure the latent state-space according to the following coupled difference equations,

$$\mathbf{z}_t^{(k)} = \mathbf{f}_k(\mathbf{z}_{t-1}^{(k)}) + \sum_{\ell \neq k} \mathbf{C}_{k,\ell}\gamma_{t-1}^{(k,\ell)} + \mathbf{G}_k\mathbf{c}_t^{(k)} + \mathbf{w}_t^{(k)} \tag{7}$$

$$\gamma_t^{(k,\ell)} = \mathbf{A}_{k,\ell}\gamma_{t-1}^{(k,\ell)} + \mathbf{B}_{k,\ell}\mathbf{z}_{t-1}^{(\ell)} \tag{8}$$

---

[1] There exist infinite state-space model realizations of minimum state dimension (Rosenbrock, 1970).

[2] Hence the lack of the $\mathbf{D}$ matrix that may be familiar in a more general treatment (Chen, 1984)

[3] 'Causal' is used in the sense of systems theory, to mean that current outputs do not depend on future inputs.

where $\mathbf{z}_t^{(k)}$ are $L_k$ dimensional region specific states, $\mathbf{w}_t^{(k)} \sim \mathcal{N}(\mathbf{0}, \mathbf{Q}_k)$, $\boldsymbol{\gamma}_t^{(k,\ell)}$ are $L_\ell M_{k,\ell}$ dimensional states of the channel from $\ell$ to $k$, $\mathbf{A}_{k,\ell}$ is $L_\ell M_{k,\ell} \times L_\ell M_{k,\ell}$, $\mathbf{B}_{k,\ell}$ is $L_\ell M_{k,\ell} \times L_\ell$ and $\mathbf{C}_{k,\ell}$ is $L_k \times L_\ell M_{k,\ell}$. We parameterize $\mathbf{f}_k(\cdot)$ as deep neural networks capable of learning highly nonlinear transition operators (specifically, we use the minimally gated unit (Heck & Salem, 2017), similar to Schimel et al. (2021)). The channel parameters, $(\mathbf{A}_{k,\ell}, \mathbf{B}_{k,\ell}, \mathbf{C}_{k,\ell})$, are learned alongside the other generative model parameters, and are constrained so that channel dynamics remain stable.

To streamline notation, we define $\boldsymbol{\Gamma}_t^{(k)}$ as an extended latent state containing the latent state vectors of the LDS that processes messages entering node $k$,

$$\boldsymbol{\Gamma}_t^{(k)} := \left( \boldsymbol{\gamma}_t^{(k,1)}, \dots, \boldsymbol{\gamma}_t^{(k,k-1)}, \boldsymbol{\gamma}_t^{(k,k+1)} \dots, \boldsymbol{\gamma}_t^{(k,K)} \right)$$

For further brevity we also define $\mathbf{s}_t^{(k)} := (\mathbf{z}_t^{(k)}, \boldsymbol{\Gamma}_t^{(k)})$, and adopt the notational convention that variables without a superscript represent the concatenation of all variables with that name,

$$\mathbf{z}_t := \left( \mathbf{z}_t^{(1)}, \dots, \mathbf{z}_t^{(K)} \right) \qquad \boldsymbol{\Gamma}_t = \left( \boldsymbol{\Gamma}_t^{(1)}, \dots, \boldsymbol{\Gamma}_t^{(K)} \right) \qquad \mathbf{s}_t = (\mathbf{z}_t, \boldsymbol{\Gamma}_t)$$

So that the full latent dynamics model, $p_{\boldsymbol{\theta}}(\mathbf{s}_t \mid \mathbf{s}_{t-1}) = \mathcal{N}(\mathbf{s}_t \mid \mathbf{m}_{\boldsymbol{\theta}}(\mathbf{s}_{t-1}), \mathbf{Q})$ factors as,

$$p_{\boldsymbol{\theta}}(\mathbf{s}_t \mid \mathbf{s}_{t-1}) = \prod_k p_{\boldsymbol{\theta}}(\mathbf{z}_t^{(k)} \mid \mathbf{z}_{t-1}^{(k)}, \boldsymbol{\Gamma}_{t-1}^{(k)}) \prod_{\ell \neq k} \delta(\boldsymbol{\gamma}_t^{(k,\ell)} \mid \boldsymbol{\gamma}_{t-1}^{(k,\ell)}, \mathbf{z}_{t-1}^{(\ell)}) \tag{9}$$

where $\delta(\cdot)$ is the Dirac delta function, and appears as a consequence of communication channel dynamics having no noise component. More than notational brevity, this representation also simplifies the algebraic complexity of developing an efficient message passing algorithm for posterior inference; since, as we discuss shortly, a state-noise inversion free algorithm can be developed, allowing us to formulate deterministic transitions as degenerate Gaussian distributions, or delta measures.

For the observation model, similar to other latent variable models of multiregion communication, each region's instantaneous activity is made dependent only on the latent variables associated with that region. This leads to a factorized likelihood, which in the linear Gaussian or Poisson GLM (generalized linear model) case we parameterize each region as,

$$p(\mathbf{y}_t^{(k)} \mid \mathbf{z}_t^{(k)}) = \mathcal{N}(\mathbf{y}_t^{(k)} \mid \mathbf{D}_k \mathbf{z}_t^{(k)} + \mathbf{d}_k, \mathbf{R}_k) \tag{10}$$

$$p(\mathbf{y}_t^{(k)} \mid \mathbf{z}_t^{(k)}) = \text{Poisson}\left( \mathbf{y}_t^{(k)} \mid \exp(\mathbf{D}_k \mathbf{z}_t^{(k)} + \mathbf{d}_k) \right) \tag{11}$$

An important concept worth noting, is that because one regions' latent variables do not *instantaneously* affect another region's activity, causal message passing algorithms should also not use observations of one region to update the filtering belief of another. For linear and Gaussian observation models, this structure arises naturally, but for amortized approximate inference (full details in Appendix D), we make sure to adhere to this principle when causally updating our filtered beliefs.

**Parameterizing channels.** The recurrent dynamics of between channel filters are parameterized as real representations of a diagonal matrix with $M$ complex-conjugate roots,

$$\mathbf{A}_{k,l} = \text{diag}\left( \begin{bmatrix} a_{k,l,1}\mathbf{I}_{L_\ell} & -b_{k,l,1}\mathbf{I}_{L_\ell} \\ b_{k,l,1}\mathbf{I}_{L_\ell} & a_{k,l,1}\mathbf{I}_{L_\ell} \end{bmatrix}, \dots, \begin{bmatrix} a_{k,l,M}\mathbf{I}_{L_\ell} & -b_{k,l,M}\mathbf{I}_{L_\ell} \\ b_{k,l,M}\mathbf{I}_{L_\ell} & a_{k,l,M}\mathbf{I}_{L_\ell} \end{bmatrix} \right) \tag{12}$$

Increasing $M$ increases the *order* of the linear filter and makes it possible to learn linear filters with increasingly nuanced frequency responses (as a result of adding additional pole-zero structures) (Stoica et al., 2005). Constraining $\mathbf{A}_{k,l}$ to be diagonal might first seem like a restrictive choice, however, with $\mathbf{B}_{k,\ell}$ and $\mathbf{C}_{k,\ell}$ free, this parameterization is able to capture any rational transfer function with greatest common denominator of order $M$ and no repeated poles (Aoki, 2013), and is the basis for Gilbert's method of constructing minimal realizations (Gilbert, 1963). Additionally, considering that diagonal matrices are dense in the space of square matrices (Golub & Van Loan, 2013), it is not possible to learn non-trivial Jordan block structures through gradient descent without enforcing those structures.

We parameterize the complex conjugate roots of each block using their representation in polar coordinates and enforce stability during optimization through clipping if a root's radius exceeds 1; however, an alternative that allows for unconstrained optimization would be the stable exponential parameterization introduced in Orvieto et al. (2023). For the readout/readin matrices, $\mathbf{C}_{k,\ell}$ and $\mathbf{B}_{k,\ell}$, their parameters are optimized without any additional constraints or structure. While we make this choice for practical simplicity, more sophisticated a priori pole-zero specifications could be introduced by considering the sparsity structure of these matrices (Kailath, 1980; Kay, 1988).

**Inference and end-to-end learning.**  While the choice of a nonlinear dynamics for local population activity is well motivated by the inability of linear dynamics models to capture key features of neural computation, such as attractor structure (Khona & Fiete, 2022), this choice also renders the exact posterior intractable – necessitating approximate inference. We approach this problem using an end-to-end variational inference methodology (Blei et al., 2017), so that gradients of the evidence lower bound (ELBO),

$$\mathcal{L}(q) = \sum_{t,k} \mathbb{E}_{q_t}\left[\log p\left(\mathbf{y}_t^{(k)} \mid \mathbf{z}_t^{(k)}\right)\right] - \mathbb{E}_{q_{t-1}}\left[\mathbb{D}_{\mathrm{KL}}\left(q(\mathbf{s}_t^{(k)} \mid \mathbf{s}_{t-1}^{(k)}) \middle\| p_{\boldsymbol{\theta}}(\mathbf{s}_t^{(k)} \mid \mathbf{s}_{t-1}^{(k)})\right)\right] \quad (13)$$

can be used to optimize the parameters of an approximation $q(\mathbf{s}_t \mid \mathbf{s}_{t-1}) \approx p(\mathbf{s}_t \mid \mathbf{s}_{t-1}, \mathbf{y}_t)$, and parameters of the generative model (derivation of the ELBO in Appendix B). A Monte-Carlo approximation of the ELBO, suitable for gradient based optimization, can be obtained by recursively sampling $\mathbf{s}_t^s \sim q(\mathbf{s}_t \mid \mathbf{s}_{t-1}^s)$ from the conditional variational approximation. For efficient sampling amenable to the reparameterization trick (Kingma & Welling, 2014), we parameterize the variational conditional as the product of a Gaussian potential, depending on $\mathbf{y}_t$, and the prior transition model by setting,

$$q(\mathbf{s}_t \mid \mathbf{s}_{t-1}) = \mathcal{N}(\mathbf{s}_t \mid \mathbf{m}_{t|t-1}, \mathbf{P}_{t|t-1}) \propto \phi(\mathbf{y}_t \mid \mathbf{s}_t) \times p_{\boldsymbol{\theta}}(\mathbf{s}_t \mid \mathbf{s}_{t-1}) \quad (14)$$

with $p_{\boldsymbol{\theta}}(\mathbf{s}_t \mid \mathbf{s}_{t-1}) = \mathcal{N}(\mathbf{s}_t \mid \mathbf{m}_{\boldsymbol{\theta}}(\mathbf{s}_{t-1}), \mathbf{Q})$. This parameterization, inspired by conjugate potential amortized inference networks such as the structured variational autoencoder (SVAE) (Johnson et al., 2016), forces the backpropagated gradients of the ELBO to traverse through the latent dynamics model – an important component for learning meaningful dynamics models capable of long horizon forecasts (Karl et al., 2017; Klushyn et al., 2021). Each Gaussian potential has the form,

$$\phi(\mathbf{y}_t \mid \mathbf{s}_t) \propto \exp\left(\mathbf{k}(\mathbf{y}_t)^\top \mathbf{s}_t + ||\mathbf{K}(\mathbf{y}_t)\mathbf{s}_t||^2\right) \quad (15)$$

When the observation model is not conjugate $\mathbf{k}(\cdot)$ and $\mathbf{K}(\cdot)$ are parameterized by neural networks whose parameters are learned maximizing the ELBO, but in the case of a linear and Gaussian observation model, $p(\mathbf{y}_t \mid \mathbf{s}_t) = \mathcal{N}(\mathbf{y}_t \mid \mathbf{D}\mathbf{s}_t, \mathbf{R})$, have optimal closed form solutions,

$$\mathbf{k}(\mathbf{y}_t) = \mathbf{D}^\top \mathbf{R}^{-1} \mathbf{y}_t \qquad\qquad \mathbf{K}(\mathbf{y}_t)\mathbf{K}(\mathbf{y}_t)^\top = \mathbf{D}^\top \mathbf{R}^{-1} \mathbf{D} \quad (16)$$

Now, given a sample $\mathbf{s}_{t-1}^s$, forming the conditional Gaussian approximation statistics,

$$\mathbf{m}_{t|t-1} = \mathbf{m}_{\boldsymbol{\theta}}(\mathbf{s}_{t-1}^s) + \mathbf{Q}\mathbf{g}_t \qquad\qquad \mathbf{P}_{t|t-1} = \mathbf{Q} - \mathbf{Q}\mathbf{K}_t(\mathbf{I} + \mathbf{K}_t^\top \mathbf{Q}\mathbf{K}_t)^{-1}\mathbf{K}_t^\top \mathbf{Q} \quad (17)$$

with $\mathbf{g}_t = \mathbf{k}_t - \mathbf{K}_t(\mathbf{I} + \mathbf{K}_t^\top \mathbf{Q}\mathbf{K}_t)^{-1}\mathbf{K}_t^\top (\mathbf{Q}\mathbf{k}_t - \mathbf{m}_{\boldsymbol{\theta}}(\mathbf{s}_{t-1}^s))$, we can sample $\mathbf{s}_t^s \sim q(\mathbf{s}_t \mid \mathbf{s}_{t-1}^s)$. Having formulated the recursive belief updates without requiring inversion of the state-noise further allows us to treat hybrid/stochastic latent transitions similarly (Appendix A). Proceeding with this recursion until time $T$ produces a completely differentiable trajectory sampled from the series of causally constructed beliefs which can be used to evaluate the ELBO. We offer a more in depth discussion of the approximate filtering algorithm in Appendix D; there, we also cover in greater detail how block structures appearing in $\mathbf{k}_t$ and $\mathbf{K}_t$ due to the multi-region observation model and deterministic channel transitions can be exploited to reduce the computational complexity of inference.

## 3  RESULTS

### 3.1  MRDS-IR RECOVERS GROUND TRUTH DYNAMICS

We first validated the efficacy of our inference algorithm by simulating a synthetic dataset with matched generative model structure (Fig. 1A). The synthetic three region system was crafted so that each area's recurrent dynamics are characterized by slow dynamic structures believed to play important roles for neural computation (Fig. 1B from left to right – a stable limit cycle produced by van der Pol's oscillator, a stable spiral, and a ring attractor), with a limited set of connections between them, themselves modeled as linear with predefined impulse response functions (Fig. 1C and D, black). From this system we generate 1000 trials of 200 time points each and project each region's latent state to a 100 dimensional observation space via a linear gaussian likelihood. We fitted a MRDS-IR model with three nodes and all-to-all connections between them, and a linear gaussian observation model to this simulated data, and assessed whether the estimated model could i) recover individual region dynamics ii) recover the linear filters between channels and iii) correctly identify whether a channel was 'open' or 'closed.'

Examining the estimated flow fields of each of the regions (Fig. 1B, bottom row), one can see that our estimator was able to learn the true autonomous dynamical systems structure, up to expected model invariances(axis rotation, and re-scaling). Fig. 1C shows the ground truth and recovered impulse

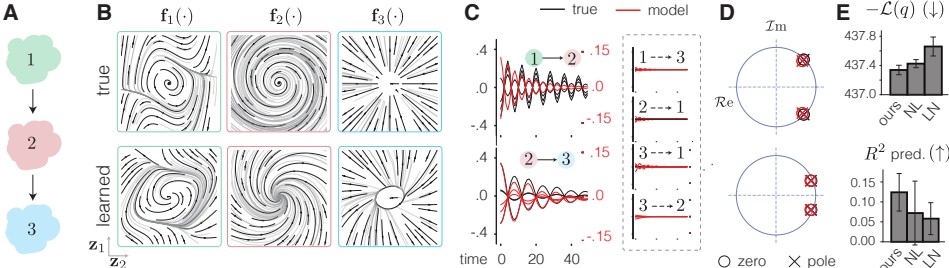

Figure 1: **Ground truth model recovery. A)** Structure of the ground truth data: three nodes and the flow of information between them. **B)** Single node population dynamics for the (top) ground truth data and (bottom) learned dynamical system after fitting the model; trajectories from the autonomous dynamics in light gray. **C)** Learned and ground truth impulse responses for the open/used (left) and closed/unused (right) channels. **D)** Pole-zero plot of the impulse response channel (black: ground truth, red: estimates). **E)** negative ELBO and predicted neural responses $R^2$ for MRDS-IR (ours) and the linear (LN) and nonlinear (NL) baseline models, see text for details.

responses of the (left) open channels and (right) closed channels. We found that the recovered impulse responses match in periodicity, although the units of amplitude are arbitrary. Importantly, the model learned to prune inactive channels by setting their amplitude close to zero.[4] The pole-zero plot in Fig. 1D confirms that the estimated active channels have frequency responses matching ground truth. Reassuringly, we found that our model –which matches the true data statistics– is revealed as the better fit in model comparison using either the ELBO or the $R^2$ of latent trajectory predictions for a forecast horizon of 150 time points regressed to ground truth examples(Fig. 1E); where the latter metric helps to assess prediction capability of the learned dynamics. This is not a given, as a simpler model could in principle fit the data better (due to finite data, fewer parameters, and a smaller inductive bias), but we confirmed that our model fitting procedure can still recover a fuller description of the underlying multi-region population activity.

## 3.2 REVERSE ENGINEERING DISTRIBUTED COMPUTATION IN AN INTEGRATION TASK

Next, we tested the ability of MRDS-IR to reveal the principles underlying multi-region neural computation in a distributed temporal integration task (Fig. 2A), which requires long time scales in the dynamics and gating of information flow between regions. Rather than engineering a multi-region dynamical system computation directly, we chose to use trained RNNs and reverse engineered their function using either the ground truth trained RNN parameters or the MRDS-IR corresponding estimates. We also included CURBD (Perich et al., 2020) as baseline comparison. Unlike the previous experiment, here there is a model mismatch between ground truth and the MRDS-IR estimator. Since CURBD uses RNNs for multi-region dynamics, it provides a particularly stringent comparison, but our explicit input conditioning which links the underlying dynamics to task meaning may help MRDS-IR to better identify the underlying computations behind the measured neural activity.

Concretely, the simulated circuit used three regions, each with a low-rank RNN architecture (Mastrogiuseppe & Ostojic, 2018; Beiran et al., 2023) so that low-dimensional dynamics could be easily visualized and compared. The activity in each region $k$ evolves as

$$\mathbf{y}_t^{(k)} = (1 - \tfrac{\Delta}{\tau_k})\mathbf{y}_{t-1}^{(k)} + \tfrac{\Delta}{\tau_k}\left(\mathbf{W}_k\phi(\mathbf{y}_{t-1}^{(k)}) + \sum_{\ell \neq k}\mathbf{W}_{k,\ell}\phi(\mathbf{y}_{t-1}^{(\ell)}) + \mathbf{G}_k\mathbf{c}_t^{(k)} + \boldsymbol{\epsilon}_t^{(k)}\right), \qquad (18)$$

where $\mathbf{c}_t^{(k)}$ is input to region $k$, read out linearly by $\mathbf{G}_k$, $\mathbf{W}_k = \mathbf{M}_k\mathbf{N}_k^\top$ and $\mathbf{W}_{k,\ell} = \mathbf{M}_{k,\ell}\mathbf{N}_{k,\ell}^\top$ are low-rank within/between population weight matrices, of ranks 1 and 2, respectively, $\boldsymbol{\epsilon}_t \sim \mathcal{N}(\mathbf{0}, \sigma^2\mathbf{I})$. Each RNN region had 128 neurons with $\tanh$ nonlinearities, and linear readouts for region-specific outputs.

The task requires each of the three nodes to integrate their respective inputs (which are constant over the course of a trial), with their computation gated by delays and region-specific go cues. Fig. 2B shows one particular example trial. At the beginning of each trial, continuous 'cue 1' and

---

[4]Loosely defined, the closed channel amplitude is much smaller than local signals in the receiving area, although this can be more precisely validated by model comparison.

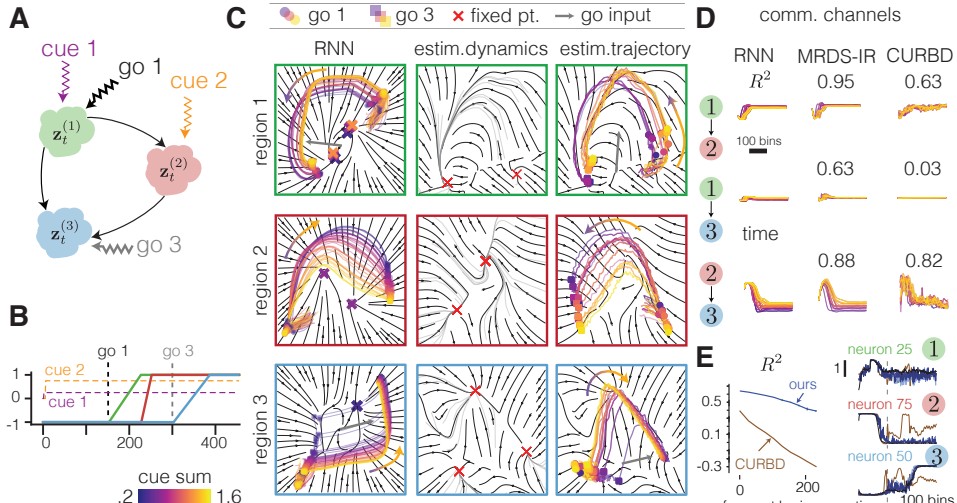

Figure 2: **A distributed working memory circuit. A)** A diagram of circuit connectivity. **B)** Structure of a single trial. Cues 1 and 2 determine speed of temporal integration; go signals start integration in the corresponding region; computation in region 2 is gated by state of region 1 with a temporal delay. **C)** Ground truth versus inferred dynamics comparison. (left) RNN autonomous dynamics and and example trajectories colored by the sum of the cues received by population 1 and 2; (middle) the dynamics learned and (right) corresponding single trajectories estimated by our MRDS-IR. **D)** Estimated inter-area communication compared to ground truth RNN signals and CURBD; numbers indicate goodness of fit measured by the $R^2$ to single trial RNN currents. **E)** Comparison of MRDS-IR and CURBD in terms of ability to predict single trial future neural responses over a forecast horizon up to 225 bins ($R^2$ of model predicted observations matched to true data, left) with one example neuron predictions for each region (right); dashed grey line marks beginning of forecasting window; black line shows ground truth neuron activity.

'cue 2' become active – with their amplitude proportional to a value, randomly chosen from the set $(10, 25, 50, 75, 90)$, indicating different speeds of integration, determining the number of time bins taken for a linear transition from '-1' to '1' of the respective region's output. The 'go 1' event (transient, binary) starts the integration process in region 1. Then, 25 time bins after that signal saturates, region 2's activity should start ramping with speed determined by 'cue 2.' Finally, when 'go 3' is received, region 3 should start ramping with a speed given by the sum of 'cue 1' and $0.8 \times$ 'cue 2.' The RNN is trained by BPTT with a supervised objective to match this target mapping.

Reflecting the two-state nature of the outputs and the working memory aspects of the task, the low-dimensional dynamics learned by the RNN (Fig. 2C, left column), have several fixed points that operate alongside the supplied input to produce the desired trajectories. Using neural activity from this RNN (all units) we then trained a three region, all-to-all connected MRDS-IR, with 2-dimensional local latent states, $L_1 = L_2 = 2$, order $M = 1$ filters for the channels, and a linear Gaussian observation model with diagonal noise. Fig. 2C (middle column) shows the phase portraits of the corresponding dynamical systems recovered by our model for each region. Similar to the ground truth RNN, the latent trajectories inferred by the MRDS-IR, Fig. 2C (right column), are driven by a combination of fixed point structures and the inputs provided by the 'go' signals. For example, focusing on Fig. 2C, the trajectories of both the RNN (left panel, bottom) and MRDS-IR (right panel, bottom), in region 3, after receiving the 'go 3' signal, are slingshot in the direction of the 'go 3' readout vector, with the autonomous dynamics then reducing their momentum according to the sum of cues for that trial.

When comparing the estimated communication to ground truth inter-region RNN currents (cue-conditioned across trial averages), we also find a good match (Fig. 2D). As their RNN counterparts, the estimated channels show ramping signals that separate across the stimuli conditions and saturate early on in the trial. Similar to the trained RNN, the relative scale of channel contents also remains in proportion; in particular, the trained RNN chose to weakly use the 1 to 3 connection, something which the estimator also picks up on. Importantly, the match to RNN communication is substantially tighter than the CURBD baseline, trained on the same data with default hyperparameters (Fig. 2D,

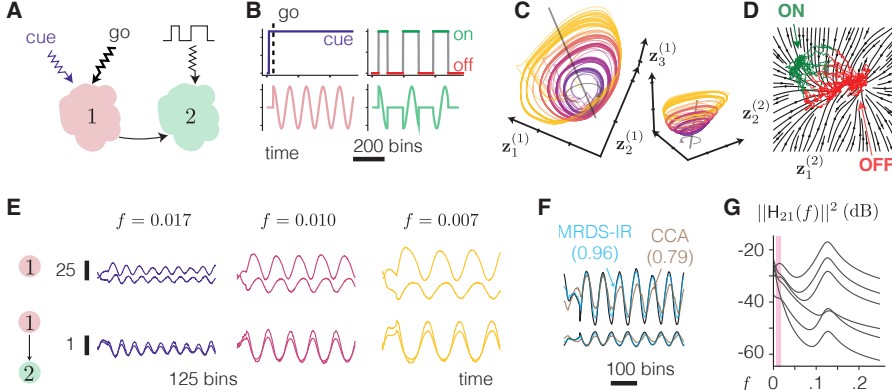

Figure 3: **Rhythmic timing task. A)** A two-node version of a rhythmic timing task: cue communicates target frequency, go the onset of the oscillation, with additional on-off switch gating outputs from region 2. **B)** Structure of an individual trial. **C)** two orientation angles showing input conditioned latent states produced by the learned autonomous dynamics of the MRDS-IR color marks different frequency cue values. **D)** Phase portrait of the dynamics for region 2 with overlayed latent trajectories, color coded by the value of the gating input. **E)** Recovered 2D latent state of region 1 (top) and the output of the channel from region 1 to region 2 (bottom) for 3 example frequency conditions. **F)** Comparison to CCA, $R^2$-measured match to ground truth above. **G)** Magnitude of the frequency response for the channel from region 1 to region 2. Each line represents one communication dimension between region 1 to region 2 (since region 1 is 3-dimensional, region 2 is 2-dimensional, and a second order filter was used, hence 6 filters total).

compare $R^2$). Moreover, the closed channels estimates are close to zero for our method, but not for CURBD (Suppl. Fig.S2), so the baseline would draw incorrect conclusions about the nature of inter-region communication in this dataset. Although it is not easy to derive the low-dimensional dynamical system structure of the multi-region dynamical system estimated by CURBD for a flow fields comparison, we can directly estimate the quality of the solution in terms of the ability of the estimated dynamics to predict neural responses (Zhao & Park, 2020; Hernandez et al., 2018) at increasingly large horizons into the future (Fig. 2E). By this metric, our approach shows better ability to match ground truth RNN dynamics across forecast horizons and a slower degradation of predictability over long time scales, reflecting a more accurate understanding of the dynamical system structure of the system.

### 3.3 A MULTIREGION RHYTHMIC TASK WITH OUTPUT GATING

While the integration task involves interesting inter-area interaction structure, it does not have a direct task equivalence to a known neuroscience experiment. For a more direct neural equivalence, we considered a variant of the rhythmic timing task introduced in Zemlianova et al. (2024), which is motivated by rhythmic timing experiments in primates (de Lafuente et al., 2022) (Fig. 3A). The oscillatory nature of the computations involves gives us an opportunity to not only expose a different form of local nonlinear dynamics than the ones considered before, but also allows us to focus on the spectral structure of the communication subspace between areas.

In the original RNN implementation of this task context/stimulus inputs indicate the desired frequency of a sinusoid (via a 'cue') to be linearly read out from the population activity after a 'go' trigger. For a multi-region variant, we assumed that the RNN passes its outputs to a second region, which aims to reproduce the same sinusoid, *except*, the second region has an 'on'-'off' switch – so its output should follow the same sinusoid region 1 produces when the switch is on, and output a value of 0 when the switch is off, as a simple form of behavioral output gating. Fig. 3B shows an example trial and the expected linear readout of the neural state for each region; each trial in the training data is chosen to have a random period ranging from 60 to 150 time bins in intervals of 15. As in the previous experiment, the RNN was trained by BPTT with each region having 128 neurons, rank 2 recurrent weights, and rank 1 cross region weights. We then fit an MRDS-IR model using latent dimensions $L_1 = 3$ and $L_2 = 2$ and order $M = 2$ channels with a linear and Gaussian observation model with independent noise for each region's 128 neurons.

In Fig. 3C, input conditioned samples from the learned dynamics model for region 1, show how the go input propels the latent state into a plane of state-space producing oscillations at a particular frequency – similar to computational mechanisms hypothesized to reproduce precise time intervals (Beiran et al., 2023; Zemlianova et al., 2024). The learned dynamics of region 2 reveal latent states for on and off switch conditions that occupy a distinct region of state-space (Fig. 3D). The communication channel from region 1 to 2 learned by MRDS-IR has clear periodic structure reflecting the target periodicity (Fig. 3E), which better matches ground truth RNN signals compared to a CCA baseline (Fig. 3F). Finally, a detailed characterization of the channel frequency response (Fig. 3G) reveals a passband at low frequencies (demarcated by the shaded region) which spans the range of frequencies that generated the data, and another passband at higher frequencies, presumably to allow for fast transitions that might occur at the time of the go signal if the gating signal is 'on.' This suggests that our method can extract interesting spectral structure for inter-area communication.

### 3.4 V1/V2 RECORDINGS

Our final experiment asked whether our method can reveal neural correlates of cross-region communication from real neurophysiological recordings. To examine this, we considered simultaneous neural recordings taken from areas V1 and V2 of a macaque monkey as it observed gratings of different orientations on a screen (Zandvakili & Kohn, 2015). This dataset has been used in other studies to examine the efficacy of intraregional models of neural signaling (Gokcen et al., 2024; Li et al., 2024), making it a natural testbed for comparison. Like Li et al. (2024) and Gokcen et al. (2024), we used spiking activity from session 106r001p26. To most directly model this data we used the Poisson GLM (generalized linear model) form of the likelihood (Weber & Pillow, 2017). To facilitate comparison, the latent space dimensionalities follow those used in Li et al. (2024), and were chosen to be $L_1 = 3$ and $L_2 = 2$, and channels were parameterized as $M = 2$ order filters, with all-to-all connectivity between regions. Since it is not clear how to best encode stimulus identity to match the biophysical circuit structure, we chose to use inputs that remained constant throughout the trial and had an amplitude proportional to the stimulus ID as a first pass.

Fig. 4A shows the stimulus-averaged latent trajectories for area V1 (top) and area V2 (bottom) extracted by the model. A key feature of these latents is their oscillatory structure, reflecting the periodicity of the drifting grating experimental stimulus. This is apparent in single trial trajectories (not shown), and remains visible in trial averages, despite potential across-trial fluctuations in phase. Moreover, different stimuli map into distinguishable regions of the latent space, reflecting the encoding of different stimuli. In Fig. 4B, we examine the V1 inferred trajectories more closely; we performed linear discriminant analysis (LDA) (Bishop, 2006) in order to find a projection that maximized the variance between trajectories with different stimulus ids and minimized variance between trajectories with the same stimulus id. The dimension of highest variance explained, shown in Fig. 4B (bottom) shows that trajectories can be separated according to their stimulus ID; while the other two dimensions, Fig. 4B (top), account for oscillations in latent space.

Investigating the interaction between the areas revealed similar oscillatory structure (Fig. 4C), qualitatively consistent with previous communication subspace based estimates (Gokcen et al., 2024). Interestingly, the amplitude of these oscillations differed markedly between feedforward and feedback information flow. As expected for sensory-driven initial communication: the V1 to V2 signal was most prominent immediately after stimulus onset, whereas the V2 to V1 communication slowly ramped up over the course of the trial, as one would expect from a top-down cognitive refinement of local information. Overall, while not in themselves surprising given the details of the experiment, the model results confirm that the extracted within area dynamics recover interpretable, known features of visual neural activity in V1/V2, which cannot be readily identified with alternative methods (see DLAG estimates in Suppl.Fig.S1).

Finally, we used the same data to quantitatively compare our approach to contemporary multi-region probabilistic methods. Specifically, we fit separately MRDS-IR, controls LN and NL (Supp.Sec. C.2), and MRM-GP (Li et al., 2024) to each of the 8 stimuli conditions. To keep the models on equal grounding, we did not make use of stimulus conditioning in MRDS-IR and used linear Gaussian observation models for each region after square-rooting the spike counts to make their statistics more Gaussian (Yu et al., 2009). Then, on a test set of 50 trials, we computed the MSE of a held-out set of neurons whose firing rate was inferred from a separate set of held-in neurons. Following a similar procedure as Li et al. (2024), we used 10 random partitions of held-in/held-out

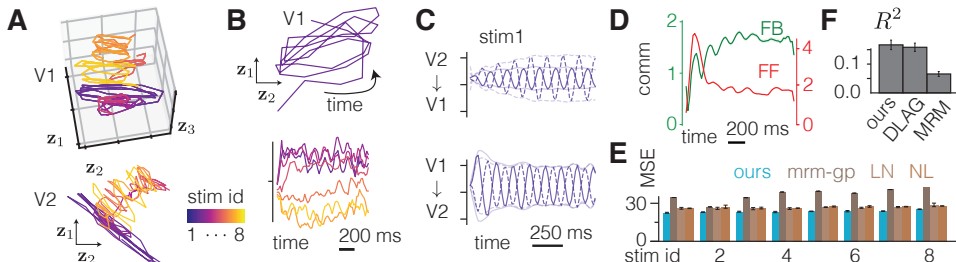

Figure 4: **Distributed computation in monkey areas V1/V2. A)** Stimulus averaged trajectories of the latent state associated with areas V1 (top) and V2 (bottom) **B)** Projection of the latent state of area V1 using LDA, where the most significant dimension (bottom) shows trajectories separate in space across time depending on the stimulus ID and (top) oscillations explained in the other two dimensions. **C)** Inferred messages transmitted across channels for one example stimulus, the darker set of lines show the two-dimensional signal transmitted while the thin line shows their spectral envelope. **D)** Signal amplitude in feedforward (FF, V1 to V2) and feedback (FB, V2 to V1) messages transmitted across channels averaged across all 8 stimulus conditions. **E)** MSE of held-out neuron predictions for each 8 stimuli for each method, averaged over 3 random seeds. **F)** $R^2$ of held-out neuron firing rate predictions (session 107l003p143, stimulus 1).

neurons, where 10% of neurons are held-out, and then reported the average across partitions. Across stimuli, MRDS-IR consistently achieved the smallest errors across models (Fig. 4E). Predicted responses on a single stimulus condition also proved competitive when compared to DLAG (Fig. 4F), using published pre-processing and $R^2$ of held-out neuron activity as a metric (Gokcen et al., 2022). Overall these results confirm that our approach is competitive to and generally outperforms state of the art methods on real multi-region data.

## 4 DISCUSSION

Despite accumulating evidence of the complex distributed nature of across-area interactions during behavior, how different aspects of the process are orchestrated across circuits remains poorly understood. To bring us closer to a process model of the distributed circuit computations that give rise to observed neural activity, we proposed a new probabilistic generative model which combines nonlinear dynamics that can capture potentially complex underlying local computations with an easy to interpret linear filtering model of inter-area communication. This unique combination of features allows for an approximate inference algorithm that uses variational principles to optimize model parameters and possible inference network parameters in an end-to-end fashion through maximization of the ELBO. Modeling channels between regions using impulse response descriptions of their input-output relationship makes it possible to use mature and principled tools from linear systems theory to understand their properties. Across a series of neurally-motivated datasets, our results demonstrate the utility of MRDS-IR in helping to determine the causal structure of multi-area computation, and recover known properties of inter-area communication in early sensory processing.

While we have allowed the model to consider all possible patterns of inter-area interactions, allowing the data to determine their relevance for explaining the observed activity, in scenarios with limited data or additional known interaction structure it may be useful to enforce additional constraints on the architecture. Future work could incorporate further regularization promoting sparse connectivity structure, for example through automatic relevance determination as in Gokcen et al. (2024), or group lasso penalties (Yuan & Lin, 2006). Another possible future direction would be exploring more structured specifications of the readout/readin matrices of the dynamics and communication channels to equip the generative model with better inductive biases.

## ACKNOWLEDGEMENTS

MD and CS were supported by the National Institutes of Health (NIH) under NIH Award R01NS127122 and NIH NINDS 1RF1NS127122-01. We thank the anonymous reviewers and area chair for their helpful feedback.

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

## A  STATE-NOISE INVERSION FREE FILTERING

To maintain a strict interpretation of channels between nodes as linear filters that deterministically process information from one region propagating to another, their recurrent dynamics do not have state-noise. For this reason, it is desirable to have a filtering algorithm that does not require inversion of the state-noise term. We accomplish this by parameterizing $q(\mathbf{z}_t \mid \mathbf{z}_{t-1}) \approx p(\mathbf{z}_t \mid \mathbf{z}_{t-1}, \mathbf{y}_t)$ as,

$$q(\mathbf{z}_t \mid \mathbf{z}_{t-1}) = \phi(\mathbf{y}_t)\, p_{\boldsymbol{\theta}}(\mathbf{z}_t \mid \mathbf{z}_{t-1}) \tag{19}$$

where $\phi(\mathbf{y}_t)$ is a Gaussian potential in terms of $\mathbf{z}_t$, so it can be written as,

$$\phi(\mathbf{y}_t) = \exp\left(\mathbf{k}_t^\top \mathbf{z}_t + ||\mathbf{K}_t \mathbf{z}_t||^2\right) \tag{20}$$

This allows sampling trajectories causally in time by exploiting conjugacy and recursively finding,

$$\mathbf{P}_{t|t-1} = \mathbf{Q} - \mathbf{Q}\mathbf{K}_t(\mathbf{I} + \mathbf{K}_t^\top \mathbf{Q}\mathbf{K}_t)^{-1}\mathbf{K}_t^\top \mathbf{Q} \tag{21}$$

$$\mathbf{m}_{t|t-1} = \mathbf{m}_{\boldsymbol{\theta}}(\mathbf{z}_{t-1}) + \mathbf{Q}\mathbf{g}_t \tag{22}$$

$$\tag{23}$$

and then drawing a sample $\mathbf{z}_t^s$ conditioned on the previous sample $\mathbf{z}_{t-1}^s$. Here, $\mathbf{g}_t$, is a quantity we define as the residual information, and is given by,

$$\mathbf{g}_t = \mathbf{k}_t - \mathbf{K}_t(\mathbf{I} + \mathbf{K}_t^\top \mathbf{Q}\mathbf{K}_t)^{-1}\mathbf{K}_t^\top(\mathbf{Q}\mathbf{k}_t - \mathbf{m}_{\boldsymbol{\theta}}(\mathbf{z}_{t-1})) \tag{24}$$

where we used the Woodbury identity to write,

$$\mathbf{m}_{t|t-1} = \mathbf{P}_{t|t-1}\left(\mathbf{Q}^{-1}\mathbf{m}_{\boldsymbol{\theta}}(\mathbf{z}_{t-1}) + \mathbf{k}_t\right) \tag{25}$$

$$= (\mathbf{Q} - \mathbf{Q}\mathbf{K}_t(\mathbf{I} + \mathbf{K}_t^\top \mathbf{Q}\mathbf{K}_t)^{-1}\mathbf{K}^\top \mathbf{Q})(\mathbf{Q}^{-1}\mathbf{m}_{\boldsymbol{\theta}}(\mathbf{z}_{t-1}) + \mathbf{k}_t) \tag{26}$$

$$= \mathbf{m}_{\boldsymbol{\theta}}(\mathbf{z}_{t-1}) + \mathbf{Q}\mathbf{g}_t \tag{27}$$

With these expressions, the KL term in the ELBO,

$$\mathbb{D}_{\mathrm{KL}}\big(q_{t|t-1}\big|\big| p_{t|t-1}\big) = \tfrac{1}{2}\left[\left((\mathbf{m}_{t|t-1} - \mathbf{m}_{\boldsymbol{\theta}}(\mathbf{z}_{t-1}))^\top \mathbf{Q}^{-1}(\mathbf{m}_{t|t-1} - \mathbf{m}_{\boldsymbol{\theta}}(\mathbf{z}_{t-1}))\right.\right. \tag{28}$$

$$\left.\left. + \operatorname{tr}(\mathbf{Q}^{-1}\mathbf{P}_{t|t-1}) + \log|\mathbf{Q}| - \log|\mathbf{P}_{t|t-1}| - L\right] \tag{29}$$

can be efficiently evaluated by rearranging and simplifying terms to find,

$$\mathbb{D}_{\mathrm{KL}}\big(q_{t|t-1}\big|\big| p_{t|t-1}\big) = \tfrac{1}{2}\left(\mathbf{g}_t^\top \mathbf{Q}\mathbf{g}_t - \operatorname{tr}(\boldsymbol{\Upsilon}\mathbf{K}^\top \mathbf{Q}\mathbf{K}\boldsymbol{\Upsilon}^\top) + \log|\boldsymbol{\Upsilon}\boldsymbol{\Upsilon}^\top|\right) \tag{30}$$

where

$$\boldsymbol{\Upsilon}\boldsymbol{\Upsilon}^\top = (\mathbf{I} + \mathbf{K}_t^\top \mathbf{Q}\mathbf{K}_t)^{-1} \tag{31}$$

## B  ELBO

We use a causal amortized variational filtering procedure that operates by drawing samples recursively from approximations, $q(\mathbf{z}_t \mid \mathbf{z}_{t-1}) \approx p(\mathbf{z}_t \mid \mathbf{z}_{t-1}, \mathbf{y}_t)$. To bound the log-marginal likelihood using causally constructed approximations of the latent state, we first bound the one-step predictive log-likelihood of an observation through standard variational arguments as,

$$\log p(\mathbf{y}_t \mid \mathbf{z}_{t-1}) \geq \mathbb{E}_{q_{t|t-1}}\left[\log p(\mathbf{y}_t \mid \mathbf{z}_t)\right] - \mathbb{D}_{\mathrm{KL}}(q(\mathbf{z}_t \mid \mathbf{z}_{t-1})|| p(\mathbf{z}_t \mid \mathbf{z}_{t-1})) \tag{32}$$

$$\coloneqq \mathcal{E}(\mathbf{y}_t \mid \mathbf{z}_{t-1}) \tag{33}$$

where the bound is tight when $q(\mathbf{z}_t \mid \mathbf{z}_{t-1}) = p(\mathbf{z}_t \mid \mathbf{y}_t, \mathbf{z}_{t-1})$. Using this bound, we proceed by applying $\log \mathbb{E}_{p(\mathbf{z}_{t-1}|\mathbf{y}_{1:t-1})}[\exp(\cdot)]$ to both sides,

$$\log \mathbb{E}_{p(\mathbf{z}_{t-1}|\mathbf{y}_{1:t-1})}\left[p(\mathbf{y}_t \mid \mathbf{z}_{t-1})\right] = \log p(\mathbf{y}_t \mid \mathbf{y}_{1:t-1}) \tag{34}$$

$$\geq \mathbb{E}_{p(\mathbf{z}_{t-1}|\mathbf{y}_{1:t-1})}\left[\mathcal{E}(\mathbf{y}_t \mid \mathbf{z}_{t-1})\right] \tag{35}$$

allowing us to construct the following lower bound on the log-marginal likelihood of $\mathbf{y}_{1:T}$ as,

$$\log p(\mathbf{y}_{1:T}) = \sum \log p(\mathbf{y}_t \mid \mathbf{y}_{1:t-1}) \tag{36}$$

$$\geq \mathbb{E}_{p(\mathbf{z}_{t-1}|\mathbf{y}_{1:t-1})}\left[\mathcal{E}(\mathbf{y}_t \mid \mathbf{z}_{t-1})\right] \tag{37}$$

This quantity depends on the intractable filtering distribution, so we approximate it using the variational approximation,

$$\log p(\mathbf{y}_{1:T}) \geq \sum \mathbb{E}_{p(\mathbf{z}_{t-1}|\mathbf{y}_{1:t-1})}\left[\mathbb{E}_{q_{t|t-1}}\left[\log p(\mathbf{y}_t \mid \mathbf{z}_t)\right] - \mathbb{D}_{\mathrm{KL}}(q(\mathbf{z}_t \mid \mathbf{z}_{t-1})|| p(\mathbf{z}_t \mid \mathbf{z}_{t-1}))\right]$$

$$\approx \sum \mathbb{E}_{q_t}\left[\log p(\mathbf{y}_t \mid \mathbf{z}_t)\right] - \mathbb{E}_{q_{t-1}}\left[\mathbb{D}_{\mathrm{KL}}(q(\mathbf{z}_t \mid \mathbf{z}_{t-1})|| p_{\boldsymbol{\theta}}(\mathbf{z}_t \mid \mathbf{z}_{t-1}))\right] \tag{38}$$

$$\coloneqq \mathcal{L}(q) \tag{39}$$

Now, although this is fundamentally an approximation to the log-marginal likelihood, we retain a strict lower bound on the data log-marginal likelihood, since, as shown in Krishnan et al. (2016),

DLAG V1/V2 communication latents

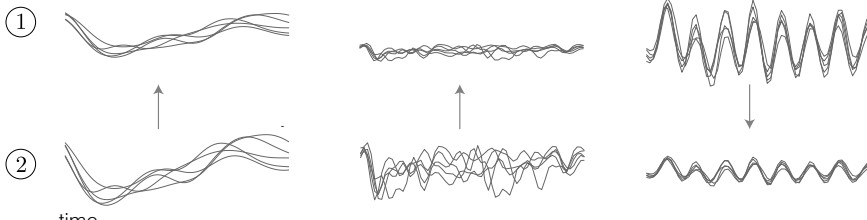

Figure S1: Example of DLAG across-region latent variables for session 107l003p143 for stimulus 1; each column represent samples of the three-dimensional across-area latent trajectory and the direction of the arrow indicates the direction of signal propagation as inferred by DLAG. These do not show a natural segregation into a FF-like early transient and a ramping up FB-like amplitude.

any variational approximation factorizing forward in time as $q(\mathbf{z}_{1:T}) = q(\mathbf{z}_1) \prod q(\mathbf{z}_t \mid \mathbf{z}_{t-1})$, can be used to lower bound $\log p(\mathbf{y}_{1:T})$ as,

$$\mathcal{F}(q) = \mathbb{E}_{q_t}\left[\log p(\mathbf{y}_t \mid \mathbf{z}_t)\right] - \mathbb{E}_{q_{t-1}}\left[\mathbb{D}_{\mathrm{KL}}(q(\mathbf{z}_t \mid \mathbf{z}_{t-1}) \| p_{\boldsymbol{\theta}}(\mathbf{z}_t \mid \mathbf{z}_{t-1}))\right] \tag{40}$$

the bound being tight when $q(\mathbf{z}_t \mid \mathbf{z}_{t-1}) = p(\mathbf{z}_t \mid \mathbf{z}_{t-1}, \mathbf{y}_{t:T})$. This means that, conditional variational approximations that are designed without using future data will lead to a looser bound. However, the benefit is that the causal structure of the generative model is enforced during inference.

## C  MULTIAREA METHODS COMPARISON

Below we quantify existing probabilistic models of multiple populations of neurons.

Table S1: Tabulating existing multipopulation models. Our model situates itself uniquely, allowing for nonlinear region specific dynamics but interpretable communication channels between regions.

| method | single node latents | communication channel | conditions on stimulus | reference |
|---|---|---|---|---|
| ours | nonlinear | impulse response | yes | |
| gLARA | LDS | linear, autoregressive | no | Semedo et al. (2014) |
| mp-srLDS | switching LDS | linear, instant | no | Glaser et al. (2020) |
| dLAG | GP | GP (time-delay) | no | Gokcen et al. (2024) |
| MRM-GP | GP | GP (frequency/phase delays) | no | Li et al. (2024) |
| mr-srLDS | switching nonlinear | switching nonlinear instantaneous | yes | Karniol-Tambour et al. (2022) |
| CURBD | RNN | linear instantaneous | no | Perich et al. (2020) |

### C.1  DLAG V1/V2 COMMUNICATION LATENTS

We wondered whether qualitative evidence of feedforward/feedback signaling similar to those revealed by MRDS-IR appeared naturally in the latent variables recovered by DLAG. In Fig. S1, we plot examples of those recovered latents suggesting those signaling characteristics are absent.

### C.2  LN/NL MODELS

To asses whether the introduction of temporal processing in the communication channels and local nonlinear dynamics actually led to substantial improvements in estimation quality over simpler model choices, we considered two simpler models in which we removed the history dependence in the channels while preserving the local nonlinear dynamics (nonlinear, NL model) and additionally swapped nonlinearities for linear local dynamics (linear, LN model; equivalent to a multi-region LDS), resulting in dynamics of the form,

$$\mathbf{z}_t^{(k)} = \mathbf{f}_k\left(\mathbf{z}_{t-1}^{(k)}\right) + \sum_{\ell \neq k} \mathbf{C}_{k,\ell}\mathbf{z}_{t-1}^{(\ell)} + \mathbf{w}_t^{(k)} \qquad \text{(NL model)}$$

$$\mathbf{z}_t^{(k)} = \mathbf{F}_k\mathbf{z}_{t-1}^{(k)} + \sum_{\ell \neq k} \mathbf{C}_{k,\ell}\mathbf{z}_{t-1}^{(\ell)} + \mathbf{w}_t^{(k)} \qquad \text{(LN model)}$$

where $\mathbf{f}_k$ and $\mathbf{F}_k$ are learnable nonlinear/linear recurrent dynamics functions.

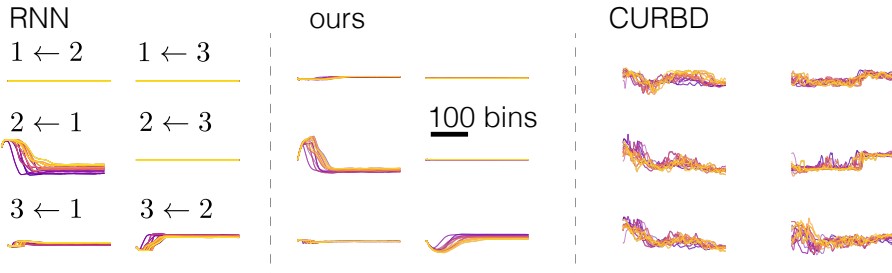

Figure S2: Inferred channels from MRDS-IR and CURBD as compared to the ground truth RNN currents. Three of the RNN inter-region connections are absent in the architecture and a forth ($3 \leftarrow$ 1) is present but weakly functional, as a result of the training procedure. Our approach recovers the right scaling for all messages, whereas CURBD infers comparable size currents for all possible pairs of regions, missing the true underlying network communication structure.

### C.3 MRDS-IR V1/V2 CHANNELS

Since the ELBO is not convex in terms of model parameters, we wondered whether the feedback-/feedforward structure of latent channels inferred by MRDS-IR was a feature that could be reproduced across different seeds. In Fig. S3 we show 8 separate seeds and the channel contents inferred by each of those models sorted left to right by decreasing ELBO values on held out data – notably, similar inferred structure is found across a majority of seeds.

### C.4 INTEGRATION TASK CHANNELS

In Fig. S2, we show all 6 channels of inferred by MRDS-IR and CURBD compared to the channels of the ground truth RNN that solve the task; examining the figure, we note that CURBD infers that the closed channels, not used by the RNN, have similar activity levels compared to the channels that were used by the RNN to solve the task.

## D FILTERING UPDATES

Recalling that,
$$\log p(\mathbf{y}_t \mid \mathbf{z}_{t-1}) \geq \sum \mathbb{E}_{q_{t|t-1}} \left[\log p(\mathbf{y}_t \mid \mathbf{z}_t)\right] - \mathbb{D}_{\mathrm{KL}}(q(\mathbf{z}_t \mid \mathbf{z}_{t-1}) \| p_{\boldsymbol{\theta}}(\mathbf{z}_t \mid \mathbf{z}_{t-1})) \tag{41}$$
if we write $p_{\boldsymbol{\theta}}(\mathbf{z}_t \mid \mathbf{z}_{t-1})$ and $q(\mathbf{z}_t \mid \mathbf{z}_{t-1})$ through their exponential family representation (Wainwright & Jordan, 2008),
$$p_{\boldsymbol{\theta}}(\mathbf{z}_t \mid \mathbf{z}_{t-1}) = h(\mathbf{z}_t) \exp \left(\boldsymbol{\lambda}_{\boldsymbol{\theta}}(\mathbf{z}_{t-1})^\top \mathbf{t}(\mathbf{z}_t) - A(\boldsymbol{\lambda}_{\boldsymbol{\theta}}(\mathbf{z}_{t-1}))\right) \tag{42}$$
$$q(\mathbf{z}_t \mid \mathbf{z}_{t-1}) = h(\mathbf{z}_t) \exp \left(\boldsymbol{\lambda}_{t|t-1}^\top \mathbf{t}(\mathbf{z}_t) - A(\boldsymbol{\lambda}_{t|t-1})\right) \tag{43}$$
where $h$ is the base measure, $A(\cdot)$ is the log-partition function, $t(\mathbf{z}_t)$ are the sufficient statistics, and $\boldsymbol{\lambda}_{\boldsymbol{\theta}}(\mathbf{z}_{t-1})$ with $\boldsymbol{\lambda}_{t|t-1}$ natural parameters of the conditional prior and approximation respectively. Then, through the Bayesian learning rule (Khan & Rue, 2023; Khan & Lin, 2017; Khan & Nielsen, 2018), the optimal variational approximation has natural parameters given by,
$$\boldsymbol{\lambda}_{t|t-1} = \boldsymbol{\lambda}_{\boldsymbol{\theta}}(\mathbf{z}_{t-1}) + \nabla_{\boldsymbol{\mu}_{t|t-1}} \mathbb{E}_{q_{t|t-1}} \left[\log p(\mathbf{y}_t \mid \mathbf{z}_t)\right] \tag{44}$$
For linear and Gaussian models with $p(\mathbf{y}_t \mid \mathbf{z}_t) = \mathcal{N}(\mathbf{D}\mathbf{z}_t, \mathbf{R})$, the RHS term can be solved for in closed form, and after some algebra we get,
$$\mathbf{P}_{t|t-1}^{-1} = \mathbf{Q}^{-1} + \mathbf{K}_t \mathbf{K}_t^\top \tag{45}$$
$$\mathbf{m}_{t|t-1} = \mathbf{P}_{t|t-1} \left(\mathbf{Q}^{-1} \mathbf{m}_{\boldsymbol{\theta}}(\mathbf{z}_{t-1}) + \mathbf{k}_t\right) \tag{46}$$
where the terms natural parameter updates, $\mathbf{k}_t$ and $\mathbf{K}_t$, are given by
$$\mathbf{k}_t = \mathbf{D}^\top \mathbf{R}^{-1} \mathbf{y}_t \tag{47}$$
$$\mathbf{K}_t \mathbf{K}_t = \mathbf{D}^\top \mathbf{R}^{-1} \mathbf{D} \tag{48}$$
However, when the observation model is nonlinear or non-Gaussian closed form updates aren't available due to the implicit nature of the solution. In this case, we amortize inference by using neural networks to parameterize conjugate potential updates as in Johnson et al. (2016); for example,

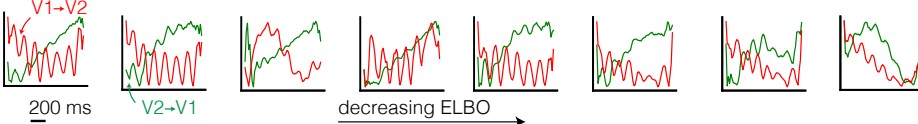

Figure S3: The stimuli averaged power content of communication channels for the area V1/V2 data across 8 different random seeds.

by directly parameterizing,

$$q(\mathbf{z}_t \mid \mathbf{z}_{t-1}) \propto \phi(\mathbf{y}_t \mid \mathbf{z}_t) \times p_{\boldsymbol{\theta}}(\mathbf{z}_t \mid \mathbf{z}_{t-1}) \tag{49}$$

where $\phi : \mathbf{y}_t \mapsto (\mathbf{k}_t, \mathbf{K}_t)$ is parameterized by a deep neural network. The parameters of the inference network can be learned alongside the generative model parameters through end-to-end learning using the ELBO objective. While for non-conjugate or non-Gaussian observations, the inference network could be parameterized in a completely black-box fashion, optionally, we can use the form of ideal implicit solutions to design a better inference network. For example, when the observation model is Poisson and $p(\mathbf{y}_t \mid \mathbf{z}_t) = \mathrm{Poisson}(\mathbf{y}_t \mid \exp(\mathbf{D}\mathbf{z}_t + \mathbf{d}))$, then the implicit and intractable optimal parameter updates are are given by,

$$\mathbf{k}_t = \mathbf{D}^\top(\mathbf{y}_t - \mathbf{r}_t - \mathrm{diag}(\mathbf{r}_t)\mathbf{D}\mathbf{m}_{\boldsymbol{\theta}}(\mathbf{z}_{t-1})) \tag{50}$$

$$\mathbf{K}_t\mathbf{K}_t^\top = \mathbf{D}^\top \mathrm{diag}(\mathbf{r}_t)\mathbf{D} \tag{51}$$

where, $\mathbf{r}_t = \exp\left(\mathbf{D}\mathbf{m}_{\boldsymbol{\theta}}(\mathbf{z}_{t-1}) + \frac{1}{2}\mathrm{diag}(\mathbf{D}\mathbf{Q}\mathbf{D}^\top) + \mathbf{d}\right)$. The way we amortize inference, inspired by these optimal updates, is to set,

$$\boldsymbol{\alpha}_t = \mathrm{NN}(\mathbf{y}_t) \tag{52}$$

$$\mathbf{r}_t = \exp\left(\boldsymbol{\alpha}_t + \mathbf{b}\right) \tag{53}$$

$$\mathbf{k}_t = \mathbf{D}^\top\left(\mathbf{y}_t - \mathbf{r}_t\right) \tag{54}$$

$$\mathbf{K}_t\mathbf{K}_t^\top = \mathbf{D}^\top \mathrm{diag}(\mathbf{r}_t)\mathbf{D} \tag{55}$$

For our experiments analyzing recordings from areas V1 and V2, we use an MLP architecture with 128 hidden units and Swish nonlinearity (Ramachandran et al., 2017) to parameterize the inference network.

Importantly, close examination of these updates shows that only latent variables read out by the likelihood will have their latent state belief updated (those unread variables can be associated with fictitious $\mathbf{0}$ columns of $\mathbf{D}$, and therefore the associated elements of $\mathbf{k}_t$ and $\mathbf{K}_t$ would be 0). In the context of multiregion models where one region's latent variables are not read out by another regions observation model, those other regions observations should play no part in updating our probabilistic belief if we causally filter data. Similarly, the belief over channel states is never updated directly through the observed data; intuitively this makes sense, showing our beliefs of the channel states can only be formed indirectly through our beliefs over the read out latent variables.

Furthemore, recalling the recursive belief updates are,

$$\mathbf{m}_{t|t-1} = \mathbf{m}_{\boldsymbol{\theta}}(\mathbf{s}_{t-1}^s) + \mathbf{Q}\mathbf{g}_t \tag{56}$$

$$\mathbf{P}_{t|t-1} = \mathbf{Q} - \mathbf{Q}\mathbf{K}_t(\mathbf{I} + \mathbf{K}_t^\top\mathbf{Q}\mathbf{K}_t)^{-1}\mathbf{K}_t^\top\mathbf{Q} \tag{57}$$

and that $\mathbf{s}_t = (\mathbf{z}_t, \boldsymbol{\Gamma}_t)$ represents the extended state, so that the blocks of $\mathbf{Q}$ associated with $\boldsymbol{\Gamma}_t$ are zero, we see that for the deterministic states their conditional covariance is $\mathbf{0}$, and $\mathbf{m}_{t|t-1} = \mathbf{m}_{\boldsymbol{\theta}}(\boldsymbol{\Gamma}_{t-1}^s)$, as expected.

**Gaussian likelihood example.** To gain some more intuition about the sparsity structure of the complete generative model, we can examine its block structure in more depth. For illustrative purposes, consider the following generative model with linear and Gaussian observations,

$$p_{\boldsymbol{\theta}}(\mathbf{s}_t \mid \mathbf{s}_{t-1}) = \prod_k p_{\boldsymbol{\theta}}(\mathbf{z}_t^{(k)} \mid \mathbf{z}_{t-1}^{(k)}, \boldsymbol{\Gamma}_{t-1}^{(k)}) \prod_{\ell \neq k} \delta(\boldsymbol{\gamma}_t^{(k,\ell)} \mid \boldsymbol{\gamma}_{t-1}^{(k,\ell)}, \mathbf{z}_{t-1}^{(\ell)}) \tag{58}$$

$$p(\mathbf{y}_t^{(k)} \mid \mathbf{z}_t^{(k)}) = \mathcal{N}(\mathbf{y}_t \mid \mathbf{D}_k\mathbf{z}_t, \mathbf{R}_k) \tag{59}$$

where again, $\mathbf{s}_t = (\mathbf{z}_t, \mathbf{\Gamma}_t)$. Now, if we construct the block matrices,

$$
\mathbf{D} = \begin{pmatrix} \mathbf{D}_1 & \mathbf{0} & \cdots & \mathbf{0} & \mathbf{0} \\ \mathbf{0} & \mathbf{D}_2 & \cdots & \mathbf{0} & \mathbf{0} \\ \vdots & \vdots & \vdots & \ddots & \vdots \\ \mathbf{0} & \mathbf{0} & \cdots & \mathbf{D}_K & \mathbf{0} \end{pmatrix} \qquad \mathbf{R} = \begin{pmatrix} \mathbf{R}_1 & \mathbf{0} & \cdots & \mathbf{0} \\ \mathbf{0} & \mathbf{R}_2 & \cdots & \mathbf{0} \\ \vdots & \vdots & \ddots & \vdots \\ \mathbf{0} & \mathbf{0} & \cdots & \mathbf{R}_K \end{pmatrix} \tag{60}
$$

where the last block column of $\mathbf{D}$ has as many columns as $\mathbf{\Gamma}_t$ has entries, then the observation model could also be written as,

$$
p(\mathbf{y}_t \mid \mathbf{s}_t) = \mathcal{N}(\mathbf{y}_t \mid \mathbf{D}\mathbf{s}_t, \mathbf{R}) \tag{61}
$$

So that natural parameter updates inherit a similar block structure,

$$
\mathbf{k}_t = \mathbf{D}^\top \mathbf{R}^{-1} \mathbf{y}_t = \begin{pmatrix} \mathbf{D}_1 \mathbf{R}_1^{-1} \mathbf{y}_t^{(1)} \\ \mathbf{D}_2 \mathbf{R}_2^{-1} \mathbf{y}_t^{(2)} \\ \vdots \\ \mathbf{D}_K \mathbf{R}_K^{-1} \mathbf{y}_t^{(K)} \\ \mathbf{0} \end{pmatrix} \qquad \mathbf{K}_t \mathbf{K}_t^\top = \mathbf{D}^\top \mathbf{R}^{-1} \mathbf{D} = \mathrm{diag} \begin{pmatrix} \mathbf{D}_1^\top \mathbf{R}_1^{-1} \mathbf{D}_1 \\ \mathbf{D}_2^\top \mathbf{R}_2^{-1} \mathbf{D}_2 \\ \vdots \\ \mathbf{D}_K^\top \mathbf{R}_K^{-1} \mathbf{D}_K \\ \mathbf{0} \end{pmatrix} \tag{62}
$$

We can determine construct similar block parameter representations of the latent dynamics by considering block operator dynamics and block state-noise, so that we can write,

$$
p_{\boldsymbol{\theta}}(\mathbf{s}_t \mid \mathbf{s}_{t-1}) = \mathcal{N}(\mathbf{s}_t \mid \mathbf{m}_{\boldsymbol{\theta}}(\mathbf{s}_{t-1}), \mathbf{Q}) \tag{63}
$$

by setting,

$$
\mathbf{m}_{\boldsymbol{\theta}}(\mathbf{s}_t) = \begin{pmatrix} \mathbf{f}_1(\mathbf{z}_t^{(1)}) + \sum_{\ell \neq 1} \mathbf{C}_{1,\ell} \boldsymbol{\gamma}_t^{(1,\ell)} \\ \mathbf{f}_2(\mathbf{z}_t^{(2)}) + \sum_{\ell \neq 2} \mathbf{C}_{2,\ell} \boldsymbol{\gamma}_t^{(2,\ell)} \\ \vdots \\ \mathbf{f}_K(\mathbf{z}_t^{(K)}) + \sum_{\ell \neq K} \mathbf{C}_{K,\ell} \\ \mathbf{A}_{1,2} \boldsymbol{\gamma}_t^{(1,2)} + \mathbf{B}_{1,2} \mathbf{z}_t^{(2)} \\ \vdots \\ \mathbf{A}_{K,K-1} \boldsymbol{\gamma}_t^{(K,K-1)} + \mathbf{B}_{K,K-1} \mathbf{z}_t^{(K-1)} \end{pmatrix} \qquad \mathbf{Q} = \mathrm{diag} \begin{pmatrix} \mathbf{Q}_1 \\ \mathbf{Q}_2 \\ \vdots \\ \mathbf{Q}_K \\ \mathbf{0} \end{pmatrix} \tag{64}
$$

Using these block representations makes it more clear that the sampling procedure used for approximate inference only requires carrying out expensive linear algebraic operations with block matrices smaller than the full latent state dimension.

