# OpenReview forum: "Nonlinear multiregion neural dynamics with parametric impulse response communication channels"
_ICLR.cc/2025/Conference — ICLR 2025 Spotlight_

### Official Review · Reviewer_TZvp · 2024-11-04

**Soundness:** 3
**Presentation:** 2
**Contribution:** 3
**Rating:** 6
**Confidence:** 3

**Summary:**

This paper introduces MRDS-IR, a probabilistic generative model for capturing the dynamics and interactions of multiple neural populations. The model combines expressive nonlinear dynamics within each population with interpretable linear communication channels between them, parameterized by their impulse response functions. The authors develop an efficient variational inference and learning algorithm that handles the model's hybrid transitions without requiring state noise inversion. Through simulations and application to real neural data, they demonstrate MRDS-IR's ability to recover meaningful characterizations of the computations within and information flow between neural populations, providing a valuable tool for understanding distributed neural computation.

**Strengths:**

- Propose a method to learn nonlinear neural population dynamics and the flow of information between regions.

- Propose an end-to-end variational methodology.

**Weaknesses:**

- The writing could be improved, as there are several areas that seem unclear to me. Please refer to the questions for specific details.

- The comparison is currently only with MRM-GP, a model with linear dynamics. A better choice would be to compare with a nonlinear model like MR-SDS, though this would require additional effort since public code for MR-SDS is not available.

**Questions:**

- In Eq.10, $\mathbf{D}_k$ and $\mathbf{R}_k$ should be explicitly defined.
- Do the red numbers (0.15, 0.0, -0.15) in Figure 1C represent the max/min amplitude of the impulse responses? Why do they differ from the black numbers (0.4, 0.0, -0.4), given that the model impulse responses and true impulse responses have similar amplitudes?
- How can we explain the difference in 1->2 communication shown in Figure 2D? The estimated impulse response appears to differ from that of the RNN.
- In Figure 3C, how should we interpret the statement, ''it is clear that the communication channel from region 1 to region 2 learned by MRDS-IRI transfers the frequency content of region 1''?
- If the amplitude of a learned impulse response between channels \(i\) and \(j\) is close to zero, does this indicate that there is no communication between these channels?

---

### Official Review · Reviewer_UHzC · 2024-11-04

**Soundness:** 4
**Presentation:** 4
**Contribution:** 4
**Rating:** 8
**Confidence:** 4

**Summary:**

This paper offers a new method for modeling multiregion neural data that combines a state space model with nonlinear local dynamics and emissions with linear, time-lagged cross-region communication channels parameterized by their impulse response. Parameterizing the communication channels in this way enables analysis of the time lagged and frequency components of communication across regions, offering high expressivity and interpretability. A structured variational inference approach is proposed for the model. The paper shows the effectiveness of the model and its interpretability through a series of simulation experiments and application to a multiregion neural data set.

**Strengths:**

This is an excellent paper with a sophisticated, well motivated technical approach, that provides a novel contribution to the literature in an important and growing area of computational neuroscience - multiregion neural data analysis. The paper is well written, and does a good job of providing scientific context and literature review, and the model and technical approach are presented in a clear, instructive way. The proposed approach in many ways combines the benefits of DLAG (2022) and MR-SDS (2024) --- the proposed parameterization of the time-delayed communication channels allows interpretable time and frequency decomposition, while maintaining the expressivity benefits of a nonlinear state space modeling approach. The structured inference approach is elegant. Simulations are well motivated and their analysis and presentation are done in a careful and accessible manner despite their complexity. The application to a previously published and analyzed dataset provides novel insights.

**Weaknesses:**

The paper is very strong, but a few notes:

- Comparisons -- the paper would benefit from additional comparisons with other time lagged communication models (DLAG) and related models with nonlinear communication (MR-SDS).

- Given the sophistication of the structured inference approach, it would be nice to show and quantify that running the fit model in generative mode yields trajectories consistent with the inferred latents.

- Figure 1 - comparing elbos across models can be problematic for a number of reasons, it would be helpful to have an interpretable measure such as r^2 on the inferred latents and reconstructed observations in the model vs NL/LN cases. Perhaps more importantly, and related to the previous point, comparing generative accuracy under these models should show a strong benefit of the additional expressivity of the full generative model; since it’s a simulation, generating more data should not be a problem.

- It may be useful to comment on the limitations of the approach’s recovery of local vs communicated dynamics. In region1 of the RNN simulation, for example, the slingshot dynamics in the true RNN are not present in the local dynamics and appear to be induced by the incoming communication. By contrast, the learned flow fields for region1 in the model include the slingshot dynamics. It may be worth commenting on this, and / or visualizing the induced communication dynamics in this case.

- Figure 3c and F: in figure 3c, 2 communication channels are shown, corresponding to the two orders of the filter, while in figure 3f, 6 dimensions are shown. The text explains that 3 and 2 dimensions are used for region 1 and 2 respectively, so 6 communication dimensions in all and an order 2 filter. It may help to include a brief summary of this in the figure caption to help readability.

- Small typos: Line 291 - missing the word “are”. Line 462 -- these latents not this latents.

**Questions:**

NA

---

### Official Review · Reviewer_SJFx · 2024-11-08

**Soundness:** 4
**Presentation:** 3
**Contribution:** 3
**Rating:** 8
**Confidence:** 4

**Summary:**

This work introduces a new method for recovering the multi-area communication patterns in task-trained RNNs and real neural recordings. The authors validate the approach in both cases, introducing a new technically solid methodology to the field.

**Strengths:**

- Simulation experiments are sound and provide strong evidence that the proposed approach is valid. Hence, this work is technically quite solid, satisfying the criteria for technical soundness.

- The topic is of broader interest to the ML and neuroscience (or NeuroAI) communities, satisfying the criteria for interest but not significance (see below).

**Weaknesses:**

I believe there are three key weaknesses, which are the main reason behind more low score. If you can address these, I am happy to revise my final decision.

## Audience problem

The paper is of broader interest to the community, but still suffers from a targeted audience problem. As a neuroscientist, I found the presentation to be too complicated for how intuitive and simple the actual method is. The mathematically heavy jargon may not help attract the biologists/experimental neuroscientists to read this paper. Yet, the paper has branded its audience as neuroscientists, which I do agree. Please see my suggestions below for how to address this.

Relatedly, several relevant citations are missing. Crucially, CURBD [1], I believe there are far more groups in neuroscience working on these problems that deserve to be cited, a few well known papers include [2-4], though this is not an exhaustive list. I trust the authors will be able to balance papers from diverse groups in their revisions.

## Missing baseline models

I would have loved to see CURBD [1], and a simple CCA based analysis [2] as baseline models that should be compared to. Former uses a generative RNN model whereas latter does not model dynamics and simply learns co-fluctuation modes. I understand the appeal of "causal" modeling, but without proof that the current model captures the actual causal structure of the brain (as was done in [3] with intervention experiments), I do not believe these claims are substantiated. They do not necessarily need to be for a publication, instead it would suffice if the authors show how their model captures new and unique insights compared to these existing models. For example, I suspect CCA will create spurious correlations between two regions that are not directly connected. As for CURBD, it assumes a particular form for the nonlinearity and will therefore likely be less expressive. It would be great if these aspects are clearly fleshed out.

## Missing new insights

Though there is an application to neural recordings (Fig. 4), I am not convinced that there is a significant insight here. Oscillations are expected by the nature of the data, as also argued by the authors. The directionality is definitely an interesting prediction, but there is no clear actionable item for the biologists to follow up. Please see [1,2] for example works that have provided such actionable items such as computational predictions regarding the global communication modes between unsuspected brain regions.

**Questions:**

I believe I understood most of the work adequately. I have some minor questions

- Is Eq. (4) correct? Should the left hand side be C \times γ?

- Is the term "neuroscientifically inspired" a rigorous description? I am not sure if I would use it in such an otherwise rigorous study.

To address my weaknesses, you can take the following actions:

- To address the audience problem, I would perhaps limit the jargon on linear system theory unless absolutely necessary to explain the findings. For example, the sentence "Here we develop a probabilistic generative model that accounts for the nonlinear nature of neural dynamics and characterizes communication between regions using channels that are parameterized by their impulse response – blending expressive nonlinear region specific dynamics with interpretable characterizations of the flow of information between regions." came out of nowhere for me and took me a while to digest. Also, minor, but there wasn't much motivation built for this method in the introduction. Perhaps, you could add more insights into what is missing in current approaches and how this modeling approach will address those gaps.

- I believe both Figs 2 and 3 should have the CCA and CURBD baselines. You can use one for CCA and another for CURBD.

- Figure 4 is interesting, and shows that the modeling approach applies to real recordings. For a score higher than borderline, I would expect the authors to show a unique biological prediction with this model, and not a post-diction that explains already existing phenomena.

Edit: I believe the authors have provided sufficient evidence to convince me that this work satisfies the criteria for publication at ICLR. Specifically, the new CURBD and CCA experiments have convinced me for the significance of this new approach and I look forward to using it in my own research.

[1] Perich, M. G., Arlt, C., Soares, S., Young, M. E., Mosher, C. P., Minxha, J., ... & Rajan, K. (2020). Inferring brain-wide interactions using data-constrained recurrent neural network models. BioRxiv, 2020-12. (and Perich, M. G., & Rajan, K. (2020). Rethinking brain-wide interactions through multi-region ‘network of networks’ models. Current opinion in neurobiology, 65, 146-151.)

[2] Ebrahimi, Sadegh, et al. "Emergent reliability in sensory cortical coding and inter-area communication." Nature 605.7911 (2022): 713-721.

[3] Vinograd, Amit, et al. "Causal evidence of a line attractor encoding an affective state." Nature 634.8035 (2024): 910-918.

[4] Musall, Simon, et al. "Pyramidal cell types drive functionally distinct cortical activity patterns during decision-making." Nature neuroscience 26.3 (2023): 495-505.

**Details Of Ethics Concerns:**

NA.

---

### Meta-Review · Area_Chair_zusx · 2024-12-17

**Metareview:**

This paper presents MRDS-IR, a probabilistic generative model designed to capture the dynamics of multi-region neural populations, incorporating expressive nonlinear dynamics within each population and interpretable linear communication channels between them, parameterized by their impulse response functions. The authors develop a variational inference and learning algorithm for this model, which can handle the hybrid transitions between nonlinear local dynamics and linear communication without requiring state noise inversion. Through a series of simulation experiments and an application to real neural data, the authors demonstrate MRDS-IR’s ability to recover meaningful characterizations of both the local computations and information flow between neural regions.

The motivation of this paper is to learn a multi-region nonlinear dynamic model where channels are parameterized by their impulse response, allowing the discovery of time delays—a feature often missing in other state-space multi-region models. However, the explanation of impulse response, transfer function, and their relationship to time delay discovery needs to be clarified. From my understanding, the transfer function results in a higher-order AR for gamma, meaning \( z^k \) depends on multiple historical time points of \( z^l \). If this is correct, it should be explicitly stated.

I also find some claims questionable, particularly regarding "causal channels." An AR model, as used here, does not inherently infer causality. The delays described in this paper, captured by multi-lag AR, differ from those captured by DLAG or MRM. For example, in Figure 4 of the V1-V2 experiment, the discovered communication signals between V1 and V2 lack a temporal shift, which contrasts with the shifts observed using DLAG.

My main concerns, however, lie in the experimental design and analysis. The choice of baseline methods is inconsistent across figures. In Figure 1, the comparison is against LN and NL; in Figure 2, it’s CURBD; and in Figure 3, it’s CCA. This ad hoc selection of baselines weakens the overall experimental rigor and gives the impression that the study is not ready for publication.

The analysis of the real neural dataset (V1-V2) is shallow. Given that DLAG dedicates an entire journal paper to analyze latent communication channels, which reveal a temporal shift between V1 and V2, there should be a discussion about why MRDS-IR does not detect such shifts. This discrepancy needs to be interpreted in the context of the latent representations of different methods. Additionally, the quantitative performance evaluation feels inconsistent, using R² and MSE with different baselines.

For example, MRM, which is more recent and has shown superior performance to DLAG in its publication, performs worse than DLAG in Figure 4D. What explains this? Is it the difference in observation models (Poisson vs. Gaussian)? Or does MRDS-IR benefit from a better latent communication model or nonlinear dynamics? These points require thorough exploration.

A deeper and more reliable analysis of real neural data is essential for a neural modeling paper like this. The V1-V2 experiment is currently neither insightful nor convincing, which significantly weakens the study.

In summary, while I appreciate the reviewers' comments and the authors' rebuttal, which have significantly improved the paper, I believe that there are still a lot of improvements the authors can make according to the comments above to strength the paper. Given the generally high level of enthusiasm, I recommend for acceptance.

**Additional Comments On Reviewer Discussion:**

During the rebuttal period, the authors made several significant revisions to their manuscript, addressing key points raised by the reviewers. They added new experiments, including comparisons with CCA and dLAG, to better demonstrate their model's performance in communication subspace estimation and recovery. They also incorporated evidence of their model's ability to make long-term predictions and generate meaningful trajectories, showing improved R2 values compared to alternative models like NL and LN. Additionally, the authors responded to concerns about controls by including more experiments to validate their method, including comparisons with other models and detailed analyses of long-horizon forecasting and communication channel estimation. These revisions strengthened the manuscript and addressed reviewer feedback effectively.

---

### Decision · Program_Chairs · 2025-01-22

Accept (Spotlight)